# An Epigenomic fingerprint of human cancers by landscape interrogation of super enhancers at the constituent level

**Xiang Liu** [1], **Nancy Gillis** [2], **Chang Jiang** [3], **Anthony McCofie** [1], **Timothy I. Shaw** [1], **Aik-Choon Tan** [4], **Bo Zhao** [5], **Lixin Wan** [3], **Derek R. Duckett** [6], **Mingxiang Teng** [1] *

1 Department of Biostatistics and Bioinformatics, Moffitt Cancer Center, Tampa, Florida, United States of America, 2 Department of Cancer Epidemiology, Moffitt Cancer Center, Tampa, Florida, United States of America, 3 Department of Molecular Oncology, Moffitt Cancer Center, Tampa, Florida, United States of America, 4 Department of Oncological Sciences, Huntsman Cancer Institute, The University of Utah, Salt Lake City, Utah, United States of America, 5 Division of Infectious Disease, Department of Medicine, Brigham and Women's Hospital and Harvard Medical School, Boston, Massachusetts, United States of America, 6 Department of Drug Discovery, Moffitt Cancer Center, Tampa, Florida, United States of America

* mingxiang.teng@moffitt.org

**Data Availability Statement:** ChIP-seq data in this study were obtained from Gene Expression Omnibus (GEO) with accession ID GSE143653. Other public data were summarized in S1 Table. All analysis code in this manuscript were documented

## Abstract

Super enhancers (SE), large genomic elements that activate transcription and drive cell identity, have been found with cancer-specific gene regulation in human cancers. Recent studies reported the importance of understanding the cooperation and function of SE internal components, i.e., the constituent enhancers (CE). However, there are no pan-cancer studies to identify cancer-specific SE signatures at the constituent level. Here, by revisiting pan-cancer SE activities with H3K27Ac ChIP-seq datasets, we report fingerprint SE signatures for 28 cancer types in the NCI-60 cell panel. We implement a mixture model to discriminate active CEs from inactive CEs by taking into consideration ChIP-seq variabilities between cancer samples and across CEs. We demonstrate that the model-based estimation of CE states provides improved functional interpretation of SE-associated regulation. We identify cancer-specific CEs by balancing their active prevalence with their capability of encoding cancer type identities. We further demonstrate that cancer-specific CEs have the strongest per-base enhancer activities in independent enhancer sequencing assays, suggesting their importance in understanding critical SE signatures. We summarize fingerprint SEs based on the cancer-specific statuses of their component CEs and build an easy-to-use R package to facilitate the query, exploration, and visualization of fingerprint SEs across cancers.

## Author summary

Super enhancers are large genomic elements comprised of multiple enhancers working together to drive gene transcription. They play a crucial role in defining cell identity and act as drivers of oncogenic gene expression in cancer cells. Characterizing cancer-specific super enhancer signatures can reveal transcriptional deregulation associated with cell

on GitHub (https://github.com/tenglab/cSEAdb_plos_code). In addition, numerical values to generate manuscript graphs and histograms were documented in S1 Data. To ensure result reproducibility, we deposited the computational framework of SE fingerprint identification for NCI-60 cancers as a protocol: https://dx.doi.org/10.17504/protocols.io.kxygx38wzg8j/v1.

**Funding:** This work was supported by NIH grants R03DE030580 (MT), R01CA262530 (DRD), R01CA255398 (LW), R01AI123420 (BZ), P30CA076292 (Biostatistics and Bioinformatics Shared Resource at Moffitt Cancer Center) and Moffitt Bio2 Pilot Grant (MT). The funders had no role in study design, data collection and analysis, decision to publish, or preparation of the manuscript.

**Competing interests:** The authors have declared that no competing interests exist.

origin and malignant transformation. Here, we generated a high-resolution fingerprint of super enhancers across 60 cancer cell lines through statistical modeling of both active and inactive components inside super enhancers. Our study revealed that cancer-specific super enhancer components are highly informative in delineating the identity of cancer cells. Our findings further revealed that cancer-specific active components exhibit stronger enhancer activities compared to non-cancer-specific components, suggesting the importance of studying the functional divergence inside super enhancers across different cancer types. Finally, we generated a database of cancer-specific super enhancer signatures for 28 cancer types with a companion computational tool to facilitate the query, exploration, and visualization of these signatures across cancers.

## Introduction

The diversity in oncogenesis mechanisms across cancers and their subtypes is largely underlain by cancer molecular profiles. Numerous molecular signatures, either pan-cancer-involved or cancer-specific, are reported in genetics and epigenetics studies. These key cancer-specific signatures oftentimes act differently across cancers, including somatic mutations, DNA methylation, and dysregulated transcription of oncogenes [1–6]. Together, these signatures provide complementary knowledge in understanding divergent oncogenic mechanisms. Super enhancers (SE), a group of large genomic elements highly corresponding to cell identities, have critical functions in cancer gene regulation [7–9]. Specifically, some SEs have dominant roles in driving tumor progression compared to other genetic signatures [9–11]. For instance, multiple SEs can significantly amplify the expression of oncogene MYC and promote tumorigenesis under different mechanisms across cancers [12,13]. Evaluating cancer-specific SE signatures thus provides critical insights toward fully dissecting the divergence of cancer mechanisms.

Pan-cancer analysis has strength in prioritizing cancer-specific signatures [1,14–18]. Currently, SE presence and absence is assessed across the NCI-60 cell panel [19] and other cells [20]; however, the binary characterization ignores variation in the SE structure, such as drifting or shrinkage of SE occupancies on the genome. These differences in effects provide important functional information, especially given the dynamic size of most SEs. Indeed, we previously showed that a significant portion of SEs are involved in structural alterations between cancers [21]. The functional effects of SEs are at least in part driven by their constituent components (i.e., constituent enhancers), which have variable contributions to SEs' overall functions [22–26]. This existing knowledge suggests an urgent need to interrogate cancer-specific SE signatures at a higher resolution by zooming into SEs at the constituent level.

In this study, we revisit SE profiles using the pan-cancer dataset (i.e., H3K27Ac ChIP-seq) of the NCI-60 cell panel [19]. To achieve a high-resolution interrogation, we focus on comparing individual constituent enhancer (CE) activities instead of the presence and absence of the whole SE. We implement a mixture model to automatically discriminate active CEs from inactive CEs which enhances the identification of cancer-specific SE signatures at the CE level. We investigate cancer-cell-specific compared to non-specific active CEs in the context of SE activity to provide insight on the functional impacts of the CEs across cancers. We generate an SE-based signature fingerprints for human cancers and build an R tool to facilitate the exploration of these signatures.

## Materials and methods

### Data acquisition

Raw H3K27ac ChIP-seq of NCI-60 human cancer cell lines were downloaded from GEO repositories with accession ID GSE143653 [19]. Quality-controlled chromatin contact files of POLR2A ChIA-PET and STARR-seq for four cancer cell lines (A549—Lung Cancer, HCT116—Colorectal Cancer, K562—Leukemia, and MCF7—Breast Cancer) were downloaded from ENCODE [27] with accession numbers (ChIA-PET: ENCFF946FGU, ENCFF246ZKR, ENCFF511QFN, and ENCFF377SXL; STARR-seq: ENCFF646OQS, ENCFF428KHI, ENCFF045TVA, and ENCFF826BPU). PRO-seq of A549 were download from ENCODE with accession number of ENCFF719NYS and ENCFF454FJE. PRO-seq of K562 were downloaded from GEO with accession number of GSM1480327 [28]. Transcription factor ChIP-seq data was downloaded from ENCODE (S1 Table). Normalized gene expression profiles for 1450 cancer cell lines were downloaded from DepMap data portal [29]. DepMap gene expression values were transformed to percentile-based (0 to 100 percentile) measurements for each gene to ease the comparison of gene expression levels across genes.

### SE profile identification and normalization

Raw H3K27Ac ChIP-seq data of the NCI-60 cancer cell lines was aligned to human genome hg38 using Bowtie2 [30], followed with peak calling by MACS2 [31]. SEs were estimated using ROSE for each ChIP-seq sample [32,33]. Peaks overlapped with gene promotors (upstream 3k bp to downstream 1k bp) and blacklist regions [34] were excluded from SE estimation. Default parameters were applied in these tools. To create a unified SE candidate list across the NCI-60 cell lines, SE regions from all samples (60 cell lines x 2 replicates = 120 samples) were merged if at least 25% width overlapping is detected between SE regions. Here, requiring a smaller overlapping window decreases the risk of merging two consecutive but independent SEs, while a larger overlapping window increases the risk of over-fragmenting the same SE into multiple separate SEs. 25% was selected as the maximum cutoff with which no obvious drop was observed for the median widths of the merged SE lists (S1 Fig). It is noted that the median width and total number of the merged SEs vary less than 0.2% and 3.4%, respectively, with the overlap cutoffs ranged between 0 to 25%. This suggests the cutoff has limited effects on the final SE list. To create a unified list of candidate CEs, CEs with at least 25% width overlapping are merged following guidelines by previous publications [35,36]. Similarly, we observed limited effects of cutoff selection on the final CE list, as the median width and total number of the merged CEs vary within 8.6% and 6.5%, respectively, with the cutoffs ranged between 0 to 25%. Other enhancers were unified with the same merging parameters for downstream normalization purpose.

In order to compare SE/enhancer activities across cancer samples, we first quantified genome-wide enhancer activities based on ChIP-seq read signals at the unified enhancer regions using featureCount [37]. A matrix of enhancer activities for all cancer samples were generated. Replicate samples were aggregated together resulting in an enhancer activity matrix for 60 cancer cell lines. Enhancer activities were normalized to adjust sequencing depths across cancer cell lines using the RLE method implemented in DESeq2 package [38]. To facilitate downstream model fitting, we replace zero signals in the normalized matrix with imputed values. In brief, for a zero signal at the enhancer $X$ in a cell line $Y$, we replaced it as the half of minimum signals of all non-zero enhancers in the cell line $Y$. We repeated this process for all zero signals at different enhancers in different cell lines. The normalized and imputed CEs' activity matrix were applied in downstream analysis.

## Mixture model to discriminate active from inactive CEs

A mixture model was developed at each CE to automatically estimate active and inactive cancer cell lines. We hypothesize each candidate CE has two states, the active and inactive. High H3K27Ac ChIP-seq signals correspond to active states while the low signals indicate inactive or weak activities. A two-component mixture model was fit to determine an optimal activity cutoff separating ChIP-seq signals between the active and inactive states.

We assumed the log2-transformed distribution for each CE activity consisted of a mixture of two normal distributions. First, to facilitate model fitting of individual CEs, we estimated the model priors based on genome-wide CEs (**S2 Fig**). Activities of genome-wide CEs were fit with global mixture models for each cell line (using R package *mixtools* [39]) to estimate the mean and standard deviations of the high and low activity mixtures. These values were averaged across cell lines to generate overall active and inactive consensuses and applied as priors in fitting models for individual CEs. CE occurrences with zero signals before imputation were excluded from the global models to ensure robust fitting. Second, we fit two-component normal-distributed mixture models for each CE using their activities across 60 cell lines. Basically, the means and standard deviations of active and inactive components for individual CEs were estimated. The probability of each activity value belonging to the active component was estimated. Expectation-Maximization algorithms was used to control model converging at no more than 1e-8 change of log-likelihood [40]. Third, for a CE activity in a fit mixture model, if its probability of belonging to the active component was larger than 0.5, we assigned this activity as active; otherwise, we assigned it as inactive.

## Cancer/Cell-specific CE/SE identification

We selected cell-specific CEs if their active or inactive prevalences are below a threshold across 60 cancer cell lines (**S3 Fig**). For a given CE, active prevalence means the frequency of being active across cell lines, while inactive prevalence means the frequency of being inactive. We determined the prevalence threshold as follows. First, for a candidate prevalence threshold between 0 and 0.5, we selected cell-specific CEs with prevalence less than this threshold and performed hierarchical clustering of 60 cancer cell lines using the activities of the selected CEs. Second, the clustering results were compared to the true cell cluster information (i.e. cancer types) using the variation of information distances [41]. Variation of information is a metric to evaluate the similarity of two clustering results for the same groups of samples. The smaller the variation of information, the more similar the two clusterings are. Third, the variation of information for all candidate prevalence thresholds between 0 to 0.5 (at a step of 0.01) were calculated. Higher prevalence thresholds resulted in more selected CEs and smaller variation of information (i.e. better clustering). To balance the clustering efficiency and the cell specificity of the selected CEs, we selected the optimal prevalence threshold (~0.21) as the one corresponding to the inflection point on the curve plotted between variation of information and prevalence cutoffs. Here, inflection point is where the slope of smoothed curve equal to -1 on the plot. Finally, CEs with a active prevalence less than 0.21 were classified as cell-specific active CEs; CEs with a inactive prevalence less than 0.21 were classified as cell-specific inactive CEs.

Next, cell-specific CEs were used to define cancer-specific CEs with the rationale that cancer-specific CEs should be present in at least two cell lines for the same cancer type. For cancer types with only one cell line in the NCI-60 panel, cell-specific CEs were selected as cancer-specific CEs. Finally, SEs containing the cancer-specific CEs were summarized as cancer-specific SEs. As a result, SEs can be cancer-specific for different cancers depending on which cancers the CE components show specificity.

### Activity evaluation of the specific and non-specific CEs

To test functional differences between cell-specific CEs and non-cell-specific CEs, we evaluated enhancer activity from independent sequencing assays including STARR-seq, PRO-seq, PRO-cap, GRO-seq, and GRO-cap (S1 Table). Four cancer cell lines with public data available were chosen for analysis: A549, K562, HCT-116, and MCF7. CEs that are specific-active, specific-inactive and non-specific-active in these cell lines were quantified for enhancer activity based on these assays. In detail, quality-controlled coverage signals of these assays were downloaded from ENCODE and GEO portals [27, 28]. Enhancer activity was quantified based on the total signal coverage at CEs using the *multiBigwigSummary* (with—binsize = 10 parameter) function from deepTools [42]. Both forward and reverse strand signals were aggregated together for nascent RNA sequencing datasets. Per-kilo-base enhancer activity was calculated using signals at CEs divided by CE widths and timed 1000.

## Results

### Super enhancer encodes epigenomic diversities across cancers

Super enhancers (SE) are involved in the cancer-specific regulation of essential genes in human cancers. The same SEs might function differently across cancers in terms of which and how gene targets are regulated [20,21,43]. To understand the SE divergence across human cancers, we revisited SE profiles for 28 cancer types based on the public H3K27Ac ChIP-seq data of the NCI-60 cancer cell lines [19]. In total, 11,100 genomic regions were identified as candidate SEs that showed significantly enriched H3K27Ac signals in at least one of the cancers (**Methods**). The widths of the SE regions ranged from 0.5Kb to 1.2Mb. The number of SEs varied dramatically across cancers (**S4 Fig**) with only a small portion (~38%) of SEs detected in more than 3 cancer types, confirming cancer-specific roles of SEs [44].

Recently, we and others demonstrated the functionally critical structures inside SEs [21,22,24], motivating the study of SE divergence across cancers at a higher resolution, i.e., at the constituent enhancer (CE) level. We applied our tool *DASE* [21] to compare SE constituent differences between pair-wise cell lines (1770 comparisons across 60 cell lines, with an average of 9530 CEs per pair and 7 CEs per SE). For a given SE, *DASE* statistically evaluated the CE patterns between cell lines by considering both the divergence between cell lines and the consistency of two replicates within each cell line. It reported differential SEs with statistical significance and differential types (e.g. activity or constituent change). On average, the majority of SEs (~72%, FDR <0.05 & at least one differential type detected) showed significant differences between pair-wise cell lines (**Fig 1A**). Such differences consist of the dismissal of a few CEs, the location drift of the SEs, and the activity discharging for the entire SE across cancers (**Fig 1B**).

To understand CE divergence across cancers, we first excluded lowly-active CEs (i.e., low H3K27Ac ChIP-seq coverage) that contribute <3% of overall activity to their SE (**Fig 1C**). We removed these CEs in downstream analysis for two reasons: 1) lowly-active CEs provide minor contribution to the nomination of SE presence as defined by main SE detection tools [32,33,45]; 2) low H3K27Ac signals may correspond to ChIP-seq artifacts due to background noise and biases. The resulted active CEs exhibit skewed distribution among cancer cell lines with more CEs active in less cancers (**Fig 1D**), consistent with cancer-specific functions of SEs. In addition, a small portion of CEs (460, or 0.95%) are consistently expressed across all cancer cell lines, suggesting their pan-cancer roles. Interestingly, the list of SEs (429, or 3.86%) containing at least one consistently expressed CE is significantly larger than the list of SEs that are shared intactly across all cancers (2, or 0.02%). This indicates that these SEs, although varied

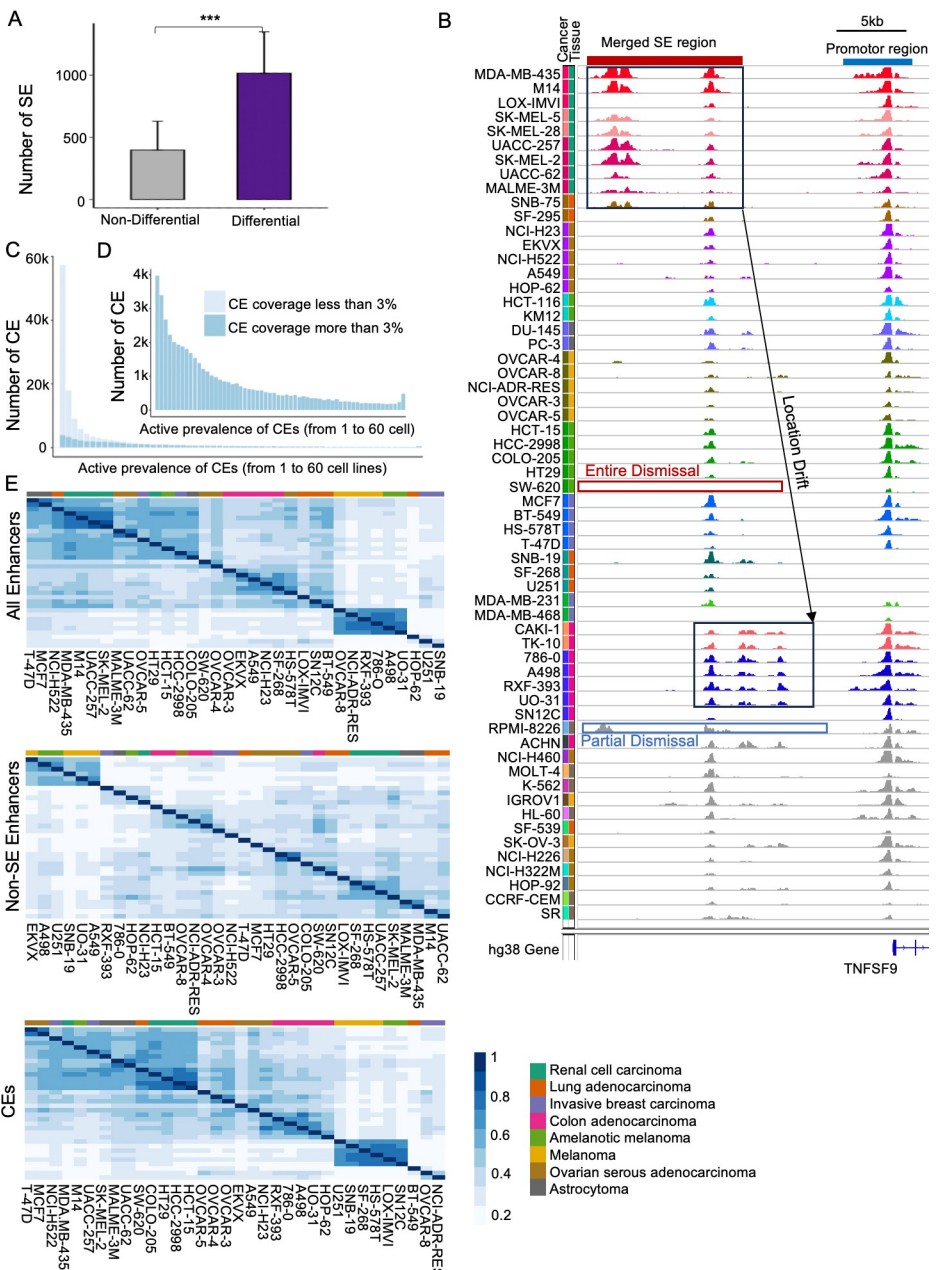

**Fig 1. Summary of SE discrepancies at the CE level across cancers. A)** Number of differential and non-differential SEs between pair-wise comparisons of NCI-60 cell lines. *Differential SEs were identified with at least one types* of differential changes *(FDR<0.05)* by *DASE* [21]. Error bars indicate standard deviation. (*** paired t-test p-value < 0.0001). **B)** An example of SE divergence across cancers involving dismissal and shifting of CEs. **C)** Frequency of CEs with different active prevalence across 60 cancer cell lines. **D)** The same as **C** but only based on CEs with high contribution to SE activity defined as H3K27Ac coverage greater than 3% of the total SE coverage. **E)** Hierarchical clustering of cancer cell lines based on the H3K27Ac-based activities of all enhancers (left), CEs with coverage greater than 3% of the SE total (middle), and non-SE enhancers (right).

partially across cancers, express strong activity at their core CEs that might be critical to cancer regulation. Dissecting the core and unique CEs across cancers will maximize our understanding of SEs' functions.

With the elevated resolution of SE activities at the CE level, we evaluated how the SE landscape can encode cancer identities. We performed unsupervised clustering using H3K27Ac ChIP-seq signals for cancers containing more than three cell lines at three different sets of genome-wide enhancers: all enhancers, the selected CEs above, and non-SE enhancers (**Fig 1E**). We estimated clustering similarities by the three enhancer sets based on alignment score [46] (lower scores mean higher similarity). Interestingly, the selected CEs show similarity or even superior clustering of cancer samples to all enhancers (alignment score of 0.009), while non-SE enhancers provide insufficient power in grouping similar cancer cells (alignment score of 0.12 to all enhancers). This indicates the capability of SEs to encode cancer identities based on their divergent CE activities. On the contrary, it also demonstrates that the non-SE enhancers exhibit heavier discrepancies in a cell-type-specific instead of cancer-specific manner. It is noted that the slight improvement of clustering by the selected CEs over all enhancers might be attributed to the exclusion of non-cancer-specific signals (the non-SE enhancers) and other noise signals (the lowly-active CEs in **Fig 1C**).

## Modeling super enhancer activities at the constituent level

To understand which CEs are involved in cancer-specific gene regulation, we compared CE activities across cancers. Previously, active CEs were defined by significant H3K27Ac ChIP-seq enrichment using peak calling algorithms for each cell line (**Fig 1D**). However, this approach might ignore critical CEs with low or marginal H3K27Ac ChIP-seq signals. First, ad hoc statistical cutoff (e.g., p-value<0.05) during peak calling brings artificial selection bias over marginal signals. Second, active consensus of enhancers by peak calling was built on genome-wide enhancer signals within a sample. It could underestimate small-size enhancers of which absolute activities are usually lower (**Fig 1B**). Third, peak calling with individual samples doesn't account for sample disparities, such as sequencing depth. A low-depth sequencing sample has less power to discriminate low CE signals from background noises (**S5 Fig**).

To provide an unbiased analysis of active CEs across cancer samples, we implemented two-component mixture models to estimate the active consensus for individual CEs (**Methods**). For a given CE region, H3K27Ac-based activity discrepancies are assumed among active samples due to reasons such as ChIP-seq measuring variability [47] (**Fig 2A**). Such discrepancies, however, should be smaller than those between the active and inactive samples. The mixture model examines the activity distribution of the CEs across all samples and converges the activities into two groups corresponding to the active and inactive states. In this case, the active consensus was built automatically based on the individual characteristics of the given CEs. We implemented one model for each CE separately. To ensure robust modeling, we normalized CE activity across the NCI-60 cell lines by scaling ChIP-seq coverage based on genome-wide enhancers (**Methods**). Zero signals were replaced with the half of sample-wise minimum signals (**Methods**).

Overall, most CEs (>99%) showed clear activity discrepancies between active and inactive cancer cell lines, while the rest of ~400 CE candidates expressed either pan-cancer activity or no activity across all samples (**Fig 2B and 2C**). The activity cutoffs to separate active and inactive samples estimated by mixture models varied across CEs (**Fig 2B**), confirming the importance of defining active CE consensus individually. Although the overall prevalence of active CEs (**S6 Fig**) does not change much from that based on peak calling (**Fig 1D**), numerous differences (> 11% of CE instances) exist in predictions of which samples these CEs are active (**Fig 2D**). 95 CE candidates with extremely low activity were identified as inactive across all samples (**Fig 2C**), suggesting they are among the false positive identifications from the peak calling method.

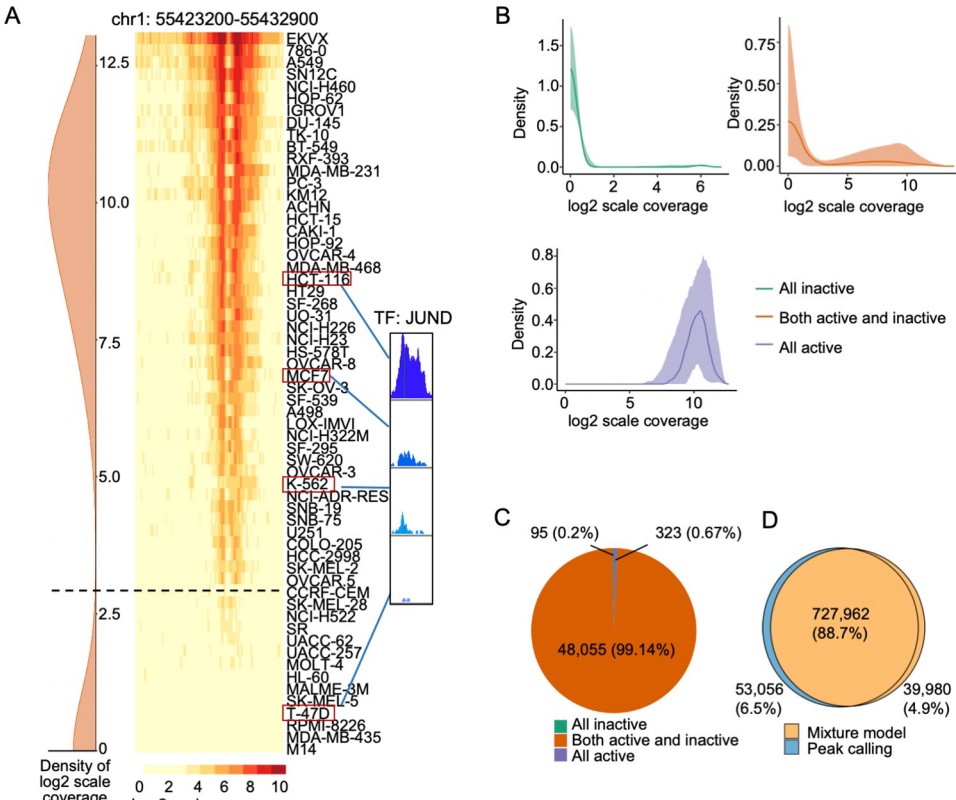

**Fig 2. Mixture models discriminating active from inactive CEs. A)** A CE example showing H3K27ac ChIP-seq coverage across 60 cell lines: left—mixture model-learned signal distribution of H3K27ac coverage; middle—coverage heatmap; right—a translation factor with binding signals correlated to the CE activity in four cell lines. **B)** Three types of CE activity densities by mixture models. Green group corresponds to CEs inactive across all cell lines; orange group corresponds to CEs active in some cell lines but inactive in others; purple group corresponds to CEs active in all cancer cell lines. The lines are the median values of the densities and the bands indicate 5% to 95% quantiles. **C)** The number of CEs belonging to each group in **B**. **D)** Comparisons of active CEs between mixture model and peak calling identification.

## Model-based states of active CEs provide improved functional interpretation

To evaluate model performance, we compared the estimated active CEs to those generated by peak calling. The active CEs by our model showed higher normalized H3K27Ac signals (log2 mean at 9.1) than those by peak calling (log2 mean at 5.1) and were comparable to the active CEs detected by both methods (**Fig 3A**). Further examination indicated the newly identified active CEs using the mixture models were those underestimated in single samples but with clear activities when compared across samples. POLR2A ChIA-PET sequencing can identify chromatin interactions between enhancers and promoters [48]. It allows us to functionally evaluate CE activities in regulating target genes. Using public POLR2A ChIA-PET datasets of four cancer cell lines [27], we found that the active CEs only by the mixture models interacted more frequently with cis-regions and linked to more target genes, compared to those by peak calling only (**Fig 3B and 3C**). More importantly, the linked target genes of active CEs by the mixture models showed higher expression levels compared to those by peak calling only (**Fig 3D**). Here, each gene's expression levels were quantified based on its expression ranking among 1450 cell lines documented in the DepMap data portal [29] (**Methods**). The strong

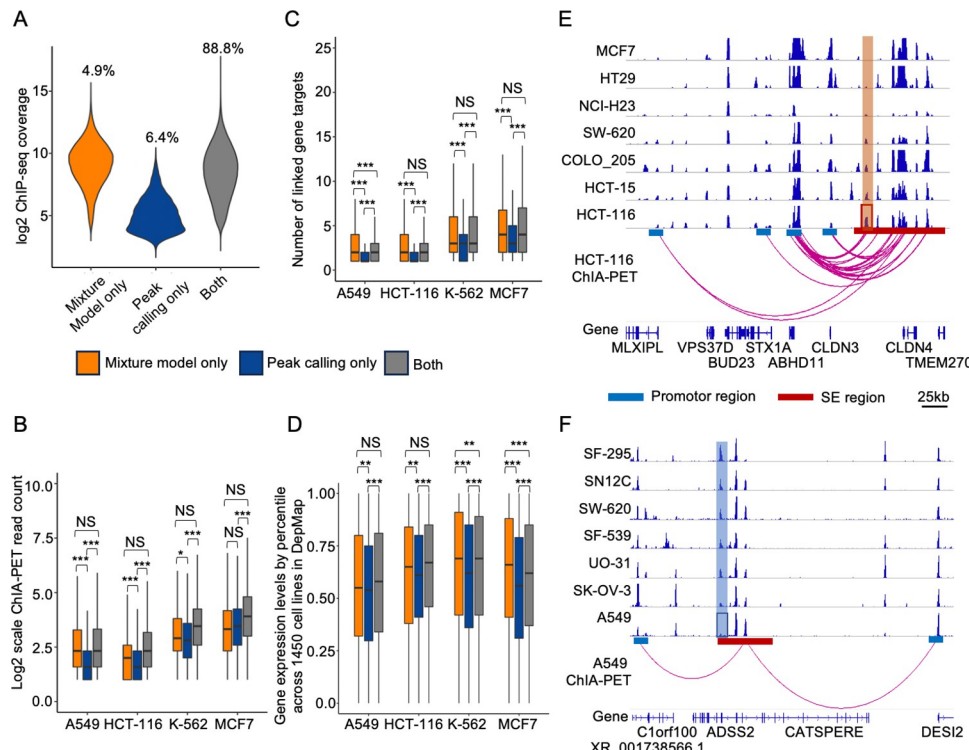

**Fig 3. Mixture-model-based CE states provide better interpretation of CE activity. A)** H3K27Ac ChIP-seq signals at active CEs predicted by different methods: only mixture model identified (orange), only peak calling identified (blue), and by both mixture model and peak calling (grey). Percentages of data points for each group are shown. **B-D)** Regulatory activities of the different groups of active CEs in **A**, as indicated by the number of associated ChIA-PET interactions (**B**), their linked targeted genes (**C**), and the expression of linked targeted genes (**D**) using ENCODE ChIA-PET datasets of four cancer cell lines. Outliers are ignored in the boxplots. Significance based on one-side Wilcoxon Rank Sum test is indicated: NS, non-significant; *, <0.05; **, <0.001; *** < 0.0001. **E-F)** Examples illustrating the improved sensitivity and specificity in detecting true active (E) and inactive (F) CEs by mixture models compared to peak calling. The red square is enhancer region under-estimated in HCT-116 (E) cell line and the blue square is enhancer region over-estimated in A549 (F) cell line by peak calling methods.

activities of CEs only by mixture models are comparable to the active CEs identified by both mixture models and peak calling. This indicates that the mixture models were capable of prioritizing the highly active enhancers for individual cell lines.

We further demonstrated the improvement of false negative/positive predictions by modeling individual CEs across samples. For example, a candidate CE was identified as inactive in the HCT-116 cell line by peak calling due to marginal activity compared to other enhancers (**Fig 3E**). However, when examined across samples, the activity is stronger than those from other cell lines at the same location. Moreover, the CE also has strong evidence linking to the target gene ABHD11 in the HCT-116 cell line as illustrated by ChIA-PET data. This indicates the CE is functionally active in the HCT-116 cell line and highlights our model's ability to rescue the false negative calls made by the standard peak calling analysis. On the contrary, **Fig 3F** showed a candidate CE that was identified as active in the A549 cell line by peak calling. However, its activity is marginal and much weaker compared to the same location in other cell lines. Moreover, it shows no evidence of linking to target genes, suggesting no functional activity in the A549 cell lines. More examples are illustrated in **S7 and S8 Figs**. Together, our systematic mixture-model-based assessment across multiple cell lines reclassified these previously inaccurate predictions of CE states.

## Cancer/cell-type-specific super enhancer signatures

We determined cancer-specific CEs by evaluating their prevalence across cancers as well as their capability to encode cancer identities. In brief, we used a variation of information [41] to estimate the grouping consistency between real cancer types and sample clustering by the selected CEs under a prevalence cutoff (**Methods**). A smaller variation of information indicates better consistency, or in other words, higher capability in encoding cancer identities by the selected CEs. A prevalence cutoff at 0.21 was chosen as the optimal to balance low variation of information and low prevalence (**Fig 4A**, **Methods**). It is noted that this threshold was also applied to select the inactive occurrences (i.e., CEs specifically lost in cancers).

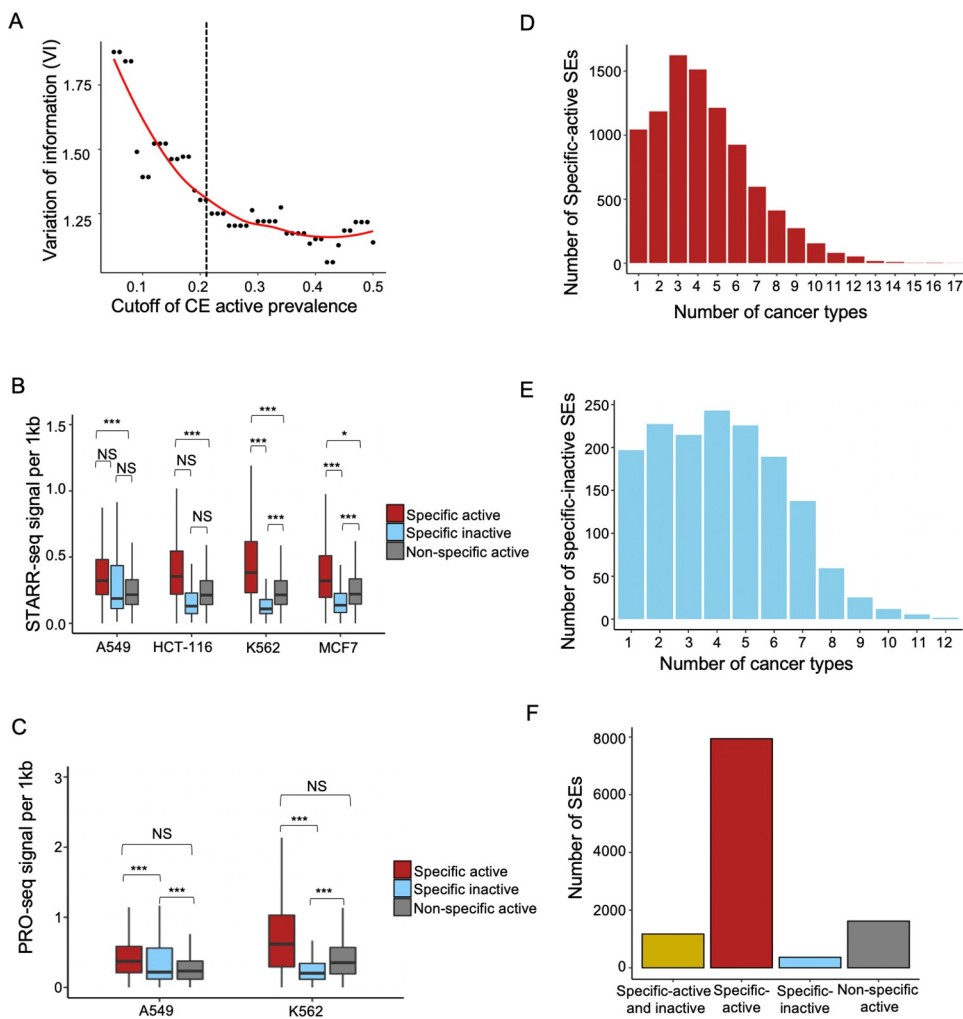

**Fig 4. Cancer/cell-specific SE signatures. A)** The capability of encoding cancer identities based on top cell-specific CEs. Running cutoffs of CE active prevalences were used to select top CEs. A cutoff of 0.21 was selected to define the final set of cell-specific CEs in downstream analysis. **B-C)** Per-kilobase enhancer activity measured by STARR-seq (B) and PRO-seq (C) for three types of CEs: cell-specific active (red); cell-specific inactive (blue) and non-specific active (grey). Significance based on student's t-test is indicated: NS, non-significant; *, $<0.05$; **, $<0.001$; *** $< 0.0001$. **D-E)** Number of SEs holding specific-active (D) or specific-inactive (E) cancer-specific CEs in different number of cancer types. Here, one SE may be counted multiple times depending on statuses of its multiple CEs. **F)** Number of SEs holding different types of cancer-specific CEs: SEs with both active and inactive cancer-specific CEs (gold); only active CEs (red); only inactive CEs (blue); SEs with no cancer-specific CEs (grey).

We evaluated the true enhancer activity of the different groups of CEs (i.e., the specific-active, specific-inactive, and non-specific-active) in four selected cell lines based on independent sequencing assays. Two main types of assays were analyzed using public data [27,28], namely the massively parallel reporter assays (STARR-seq, **Fig 4B**) and nascent RNA sequencing (Pro-seq etc., **Figs 4C and S9**). STARR-seq is a reporter-gene-based assay implemented by constructing plasmid libraries that are introduced in cells to evaluate enhancer activity at millions of candidate DNA sequences [49]. Nascent RNA sequencing can evaluate enhancer activities by measuring the local transcriptional programs at enhancers [50–52]. Normalized sequencing signals were downloaded and quantified at individual CEs for the selected cell lines (**Methods**). The specific-active CEs showed the highest per-base activity while the specific-inactive CEs presented the lowest, and the observations were reproducible between STARR-seq and Pro-seq (**Fig 4B and 4C**). This suggests the critical regulatory roles of the specific-active SEs in the corresponding cell lines. Of note, we did not observe similar differences between the three CE groups when CE width was not justified (**S10 Fig**), in which case CE activity was confounded by the genomic sizes of CEs and wider CEs were overestimated by sequencing signals.

We further considered cancer-specific CEs as those specified in at least two of the cell lines for a given cancer type. We examined which SEs contain these specific CEs, both active and inactive. Overall, the number of SEs containing specific-active CEs (9114) is larger than the number of SEs holding specific-inactive CEs (1539) (**Fig 4D and 4E**). 1173 SEs hold specific-active and specific-inactive CEs simultaneously (**Fig 4F**), suggesting their versatile roles in different cancers. Moreover, 8,387 SEs contain multiple cancer-specific CEs that show activeness in different cancers (more than 2 cancer types). In other words, these SE are cancer-specific in multiple cancers according to the statuses of its different CE components. Together, the majority of SEs contain cancer-specific CEs, suggesting improved sensitivity of our methods in interrogating cancer-specific SE signatures.

## A tool to visualize SE fingerprints across cancers

We summarized cancer-specific SEs based on their CE statuses for all cancer types in the NCI-60 cell panel (**Fig 5A**). As illustrated (**Figs 1E** and **4A**), these SEs are highly informative and can act as fingerprints in encoding cancer identities. We further implemented an R package *cSEAdb (https://github.com/tenglab/cSEAdb)*, which facilitates the query, exploration, and visualization of these SE fingerprints. The R package allows queries of cancer-specific SEs using four types of information: cancer types, cancer cell types, gene targets of SEs, or SE locations. For a query by cancer or cell types, the package returns the list of cancer-specific active and inactive SEs as well as an overview of fingerprint SEs across the genome (**Fig 5B and 5C**). For a query by gene targets, any nearby SE in at least one of the NCI-60 cells will be returned. Further zooming into individual SEs is allowed based on all query types above. When zooming into an SE or querying by an SE location, the details of SE activity are visualized with color-coded highlights of its cancer-specific CE components across cancers (**Fig 5D**).

## Discussion

In this study, we systematically investigated super enhancer (SE) profiles across human cancers based on NCI-60 cancer cell lines and generated cancer-specific SE signatures at the resolution of constituent enhancers (CE). Through computational modeling of the varied CE activities across cell lines and individual CEs, we provided an improved estimation of active CEs for cancers. We evaluated the estimations using multiple functional assays associated with enhancers, including POLR2A ChIA-PET, STARR-seq, PRO-seq etc., and demonstrated that the modeling process improves the functional interpretation of SE activity in cancers. We further

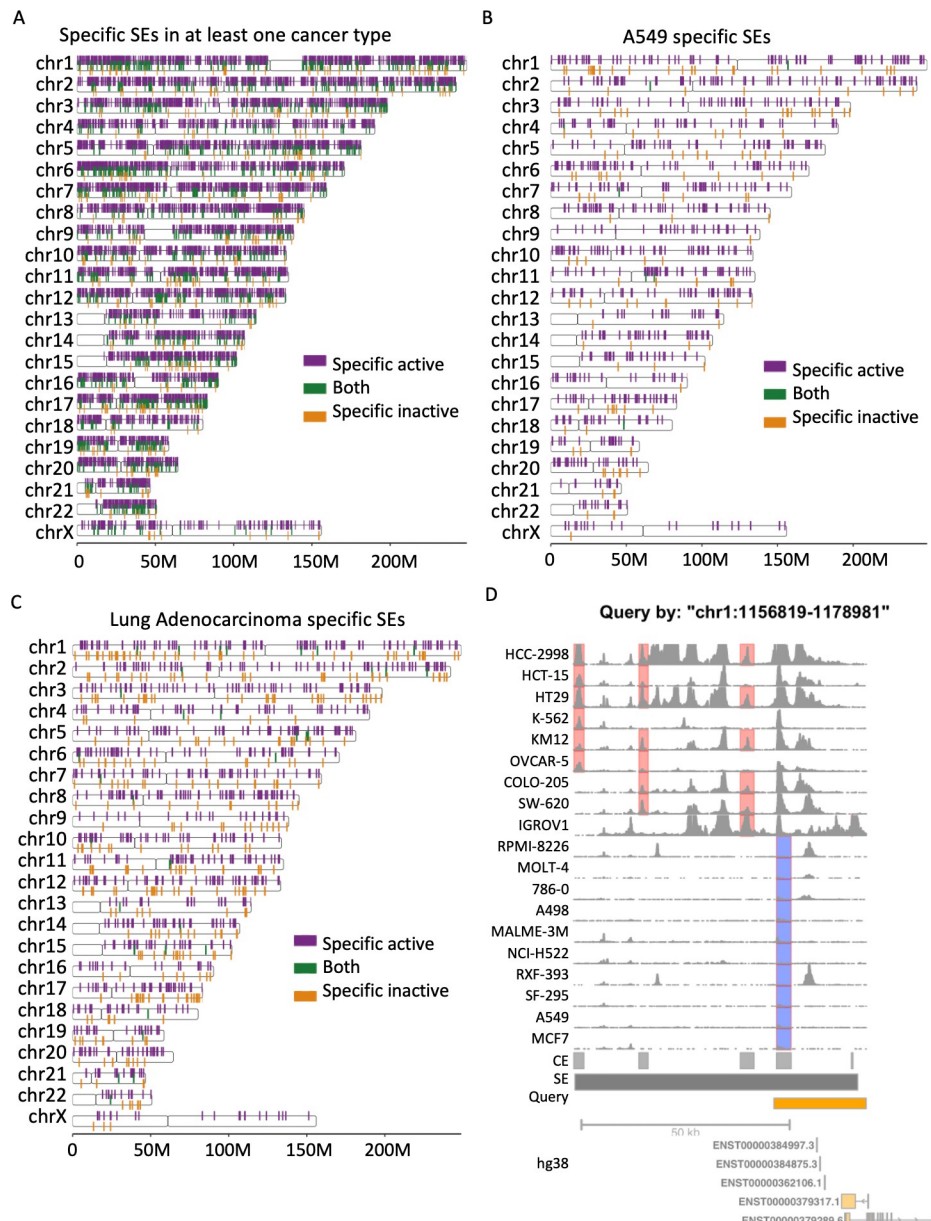

**Fig 5. A tool for exploring fingerprint SE signatures. A)** Genome-wide SE landscape with cancer-specific CEs across all studied cancer types. **B)** Genome-wide SE landscape with cell-specific CEs in A549 cell line. **C)** Genome-wide SE landscape with cancer-specific CEs in lung adenocarcinoma. **D)** H3K27Ac ChIP-seq signals for a SE example with cancer-specific active (red) and inactive (blue) CEs highlighted.

showed that cancer-specific SE signatures encode cancer identities and have uniquely strong per-base activity in the corresponding cancers. To facilitate community exploration of our identified SE signatures, we built an easy-to-use R package to document, query and visualize SEs in cancers. Together, our algorithms provide a novel insight to model and interpret cancer-specific fingerprint SE signatures (**S11 Fig**).

It is interesting to observe that cell-specific CEs exhibit the strongest per-base instead of whole-enhancer activity. On one hand, this suggests the imperative evidence that cell-specific CEs are among the key regulators in the corresponding cell lines. Generating cancer-specific

CEs can provide insights to the essential epigenomic signatures in cancers. On the other hand, it highlights the importance of unravelling CE activity with its genomic size. Enhancer regions are usually involved with multiple transcription factor binding. Enhancer size is likely reflecting the complexity of local collaborations between transcription factors instead of regulatory activity [53, 54]. The per-base activity thus may better represent the true regulatory activity induced by key transcription factors.

Cell line models may exhibit different SE profiles compared to cancer tissues [43], introducing a possible challenge to the translation of these findings. However, cancer tissues are usually heterogeneous, containing immune cells, normal cells and cancer clones at different progression stages. A future application of this work will be comparison of these cell line-based SE profiles to the deconvoluted tumor SE profiles. However, this is challenging given the limited data that exists on tissue-based single-cell-type enhancer activity. One possible first step may be use of the recent single cell CUT&Tag experiments data [55–57]. Here, we optimized the utility of our findings by use of multiple cell lines of the same cancer types when building our cancer-specific SE signatures, thus providing an additional layer of robustness toward reproducible cancer-specific signatures for a given cancer type. It is noted that only one cell line is available for half of the cancer types in the NCI-60 panel, cancer-specific signatures from these cancers might confound with cell-specific or tissue-specific signatures. Our next plan is to add more batch-effect controlled data from other cancer and normal tissue cell lines to improve our fingerprint signatures.

Our study is the first to provide cancer-specific-inactive signatures across cancers. Interestingly, we observed that their frequency is lower than the cancer-specific-active signatures. More precise interpretation of the functional impact of these signatures is currently challenging, but we suspect that the specific disappearances of these enhancers may correspond to missing activities of transcription factors at these loci. Further investigations are needed to fully understand the function of these cancer-specific lost signatures. In addition, we anticipate that future functional annotation of both specific active and inactive CEs will promote the use of the presented resource here by a broader community of the field. For instance, the highlighted CE in S8B Fig is actually a cell-specific active signature in the K562 cell line (**S12 Fig**). Compared to A549 cells, this CE exhibits unique regulation to target genes ANKRD9 and RCOR1 in K562 cells. Full characterization of such unique regulation will lead to a better understanding of SE-involved cancer mechanisms.

The proposed mixture model is robust in identifying the few uniformly inactive CE elements that were classified as active by peak calling. These CEs showed extremely weak activities with no linking to any genes based on ChIA-PET, suggesting the accuracy of our mixture models. We identified pan-cancer active core CEs for all cancers, which may provide critical insight into functional knowledge of routine cancer cell maintenance by SE-associated regulation. Although beyond the scope of this study, our mixture model can also be applied to estimate the non-SE enhancer activity states across samples.

In summary, our pan-cancer analysis demonstrates improved accuracy in identifying active CEs and elevated sensitivity in detecting cancer-specific SE signatures. The R package will facilitate the cancer-specific exploration of potential therapeutic targets in epigenomics.

## Supporting information

**S1 Fig. Median width and total number of merged SEs and CEs based on different overlap cutoffs.** Dashed line indicates 25% overlap cutoff. a. SE median width b. Total number of SEs c. CE median width d. Total number of CEs.
(PDF)

**S2 Fig. Genome-wide priors of the inactive and active enhancer groups estimated across 60 cancer cell lines.** Data points indicate the estimated means of the lower and higher mixtures in individual cell lines.
(PDF)

**S3 Fig. Flowchart to define cell/cancer-specific CEs.**
(PDF)

**S4 Fig. Number of SEs identified in different cancers.**
(PDF)

**S5 Fig. Vertical comparison of H3K27Ac signals across cancers refines enhancer activity consensus in individual samples.** Left: normalized ChIP-seq signals recovers enhancer activity in MDA-MB-231 cell line; right: raw signals indicate no active enhancers at the same location due to low ChIP-seq coverage in MDA-MB-231.
(PDF)

**S6 Fig. Overall prevalence of active CEs identified by mixture models and peak calling.**
(PDF)

**S7 Fig. Extra examples illustrating the improved sensitivity in detecting true active CEs by mixture models compared to peak calling.** (a). Red shaded square is an enhancer region under-estimated in A549 cell line by peak calling methods but identified as active by mixture model. This region shows strong enhancer activity in A549 compared to other cancer cell lines and presents regulatory interactions with target gene TRMT5 based on ChIA-PET data. (b). Similar to a) but for another enhancer region regulating SIX4 gene in A549 cell line. (c). Similar to a) but for another enhancer region regulating GNAI3 gene in HCT-116 cell line. (d). Similar to a) but for another enhancer region regulating HID1, MRPL58 and KCTD2 in HCT-116 cell line.
(PDF)

**S8 Fig. Extra examples illustrating the improved specificity in detecting true inactive CEs by mixture models compared to peak calling.** Blue shaded square is an enhancer region over-estimated in A549 cell line by peak calling methods but identified as inactive by mixture model. This region shows weak enhancer activity in A549 compared to other cancer cell lines and presents no regulatory interactions with any genes based on ChIA-PET data. (b). Similar to a) but for another enhancer region in A549 cell line. (c). Similar to a) but for another enhancer region in HCT-116 cell line. (d). Similar to a) but for another enhancer region in MCF7 cell line.
(PDF)

**S9 Fig. Enhancer activity from extra nascent RNA sequencing datasets for K562, including GRO- cap, Pro-cap and GRO-seq, from the ENCODE and GEO data repositories.**
(PDF)

**S10 Fig. STARR-seq and PRO-seq based CE activity without normalization of CE width.** a. STARR-seq. b. PRO-seq.
(PDF)

**S11 Fig. Flowchart of identifying fingerprint SE signatures across cancers.**
(PDF)

**S12 Fig. An example of interpreting cancer-specific active CEs.** Highlighted in blue square is a cell- specific active CE in K562 but inactive in A549. This CE links to promoters of two

gene targets, ANKRD9 and RCOR1, suggesting its cell-specific regulation in K562.
(PDF)

**S1 Table. Accession IDs and meta information for the publicly downloaded data in this manuscript.**
(XLSX)

**S1 Data. Numberical values to generate manuscript graphs and histograms.**
(XLSX)

## Author Contributions

**Conceptualization:** Derek R. Duckett, Mingxiang Teng.

**Formal analysis:** Xiang Liu.

**Funding acquisition:** Bo Zhao, Lixin Wan, Derek R. Duckett.

**Methodology:** Xiang Liu, Nancy Gillis, Chang Jiang, Anthony McCofie, Timothy I. Shaw, Mingxiang Teng.

**Software:** Xiang Liu, Anthony McCofie.

**Supervision:** Aik-Choon Tan, Bo Zhao, Lixin Wan, Derek R. Duckett, Mingxiang Teng.

**Writing – original draft:** Xiang Liu, Mingxiang Teng.

**Writing – review & editing:** Xiang Liu, Nancy Gillis, Chang Jiang, Timothy I. Shaw, Aik-Choon Tan, Bo Zhao, Lixin Wan, Derek R. Duckett, Mingxiang Teng.

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
