## [Decision Letter · Decision Letter 0]

4 Nov 2023

Dear Dr. Teng,

Thank you very much for submitting your manuscript "An Epigenomic Fingerprint of Human Cancers by Landscape Interrogation of Super Enhancers at The Constituent Level" for consideration at PLOS Computational Biology.

As with all papers reviewed by the journal, your manuscript was reviewed by members of the editorial board and by several independent reviewers. In light of the reviews (below this email), we would like to invite the resubmission of a significantly-revised version that takes into account the reviewers' comments.

We cannot make any decision about publication until we have seen the revised manuscript and your response to the reviewers' comments. Your revised manuscript is also likely to be sent to reviewers for further evaluation.

Sincerely,

Ilya Ioshikhes

Section Editor

PLOS Computational Biology

Ilya Ioshikhes

Section Editor

PLOS Computational Biology

Reviewer's Responses to Questions

**Comments to the Authors:**

Reviewer #1: In this manuscript, Liu et al. describe an advanced computation study focused on SEs’ structure and function in cancer. The authors applied bioinformatics analysis in diverse sets of publicly available datasets in order to decipher how SEs are structured and how their composites operate in-between diverse cancer tissue types. Overall, the study is well-designed and efficiently implemented. The authors managed to zoom into SEs entities at the constituent level, and this is an achievement. They applied straightforward criteria to distinguish active versus inactive states of SEs’ constituents, and to classify cell-type specific SEs. Finally, the authors developed a computational package that can be applied for the exploration of these findings.

The manuscript is well-written.

A number of points outlined below should be addressed:

Major Points

[1] Page 11

“POLR2A ChIA-PET sequencing can identify chromatin interactions between enhancers and promoters, thus allows us to functionally evaluate CE activity in regulating target genes. Using public POLR2A ChIA-PET datasets of four cancer cell lines, we found that the active CEs only by the mixture models interacted more frequently with cis-regions (on average 3.4 interactions) and linked to more target genes (on average 5.5 genes), compared to those by peak calling only (on average 2.9 interactions and 3.3 genes, respectively) (Fig 3B-C).”

The fact that a cis-acting element develops 3D associations in vivo with another genomic loci, e.g. a promoter or a gene, does not necessarily equal to its increased capability for being active but elevates its potential for such a functional conductance. I believe that the authors have to complement this analysis by providing details relative to the transcriptional states of the interacting genes with the CEs identified. They have to analyze gene expression data and to demonstrate any correlation between the CEs and the genes that are detected in physical proximity according to POLR2A ChIA-PET.

[2] Figure 3E: The panel demonstrates a promoter region that is located internally within the MIR1252 locus (bottom of the panel).

Is this a cryptic promoter? The authors need to clarify.

In addition, more examples analogous to those provided are required to support the findings described in this paragraph.

[3] Figure 5A and B: The resolution of the figure should be improved. The color code is not displayed in high resolution and the size of the panels should be increased in order to facilitate the readers to follow the information incorporated.

[4] Page 12

The authors have to provide more details in the main text regarding how they assessed the STARR-seq datasets. In addition, they have to highlight that STARR-seq is a reporter-gene-based assay implemented by the construction of plasmid libraries that are introduced in cells.

Minor points

[1] The authors utilize a spectrum of publicly available datasets. It will be helpful for the readers if the authors include the citations in the corresponding paragraphs of the results section.

[2] “We generated cancer-specific SEs based on their CE statuses for all cancer types in the NCI-60 cell panel (Fig 5A).”

The authors have to rephrase this sentence. I believe the verb “generate” is not the right one here.

Reviewer #2: This manuscript presents a comprehensive exploration of cancer-specific super enhancer (SE) signatures at the constituent level for 28 cancer types in the NCI-60 cell panel. The authors employ a Gaussian mixture model to distinguish active constituent enhancers (CEs) from inactive ones. Additionally, they develop an R package to facilitate data exploration and visualization. The manuscript is well written, with clear structure, and offers valuable contributions to the field. However, there are several concerns, primarily pertaining to the methodology and clarity in certain sections.

Major:

1. The description of SE and CE candidate list creation lacks clarity, especially regarding the estimation of CEs. Further elaboration on the distinction between the estimation of SEs and CEs is necessary. Additionally, a rationale for the chosen percentage thresholds (20% width overlapping for SEs and 10% width overlapping for CEs) is needed, along with relevant references to support this choice.

2. The approach to bootstrapping raises concerns. Bootstrapping typically involves creating new samples of the same size as the original by sampling with replacement. The authors' approach, generating one new sample (2.5k data points) much larger than the original (60 data points) by sampling from the estimated density function, is not appropriate. Additionally, instead of fitting a model to the data generated from a smoothened density function, fitting the mixture model directly to the original data should be considered.

3. The cause of zero values in the resulting normalized activity matrix for CEs requires clarification. Treating these zero values as missing data and applying appropriate imputation techniques may be a viable solution to this issue, obviating the need for incorrect bootstrapping strategies.

4. Figure 1A requires additional information. The methodology for conducting pairwise comparisons of SEs among the 60 cancer cell lines needs to be clarified. Details on the statistical tests employed, considering the limited replicates for each cell line, should be provided. Additionally, the number of identified CEs and the average number of CEs per SE may also be provided to improve clarity.

5. The section "Cancer/Cell-specific CE/SE identification" is challenging to comprehend. A plot illustrating this process could greatly enhance understanding. Additionally, the authors should provide further explanation and context for terms such as "prevalence threshold," "variation of information," and "inflection point."

6. Figure 2D demonstrates comparable outcomes between the mixture model-based approach and the traditional peak calling method. Given the statistical concerns with the mixture model approach and the similarity in results, it may be more beneficial to shift the focus towards the broader significance of investigating cancer-specific SEs at the constituent level. In my opinion, this constitutes a pivotal contribution, and the subsequent creation of the database (R package) presents a valuable resource for researchers in the field. Expanding on the functionality and features of the database would greatly enhance the impact of this study. This aspect merits further elaboration and emphasis in the manuscript.

7. To enhance the reproducibility of the research, I recommend that the authors provide access to the code used for processing and normalizing ChIP-seq data, implementing the mixture model, identifying cancer/cell-specific SEs and CEs, and analyzing the data generated from independent sequencing assays. Sharing this code will not only allow for validation of the methodology but also empower other researchers to apply and build upon this valuable work. This practice aligns with best practices in computational biology and will significantly benefit the scientific community.

Minor:

1. Page 8: "shewed distribution" should be corrected to "skewed distribution."

2. Page 9: The last sentence on this page contains a grammatical error and requires revision.

3. Page 11: "POLR2A ChIA-PET sequencing ... thus allows us to ..." contains a grammatical mistake and should be revised.

4. Page 16: "is demonstrated with" should be changed to "demonstrates."

5. Data availability: The word "data" is plural; please ensure that the plural form of the verb is used.

6. Figure S1: "activate group" should be corrected to "active group."

Reviewer #3: The authors summarize fingerprint SEs based on the cancer-specific statuses of their component active enhancers. Interestingly, they build an easy-to use R package to facilitate the query, exploration, and visualization of fingerprint SEs across cancers. The package will help us to get the specific epigenetic status of cancers and obtain more useful information about cancer biomarker.

I suggest the author list the code how to obtain the specific SE in pancancer.

I have no questions about other issue.

**Have the authors made all data and (if applicable) computational code underlying the findings in their manuscript fully available?**

Reviewer #1: Yes

Reviewer #2: **No: **I recommend that the authors provide access to the code used for processing and normalizing ChIP-seq data, implementing the mixture model, identifying cancer/cell-specific SEs and CEs, and analyzing the data generated from independent sequencing assays.

Reviewer #3: Yes

PLOS authors have the option to publish the peer review history of their article (what does this mean?). If published, this will include your full peer review and any attached files.

Reviewer #1: No

Reviewer #2: No

Reviewer #3: No
---

## [Editor Report · Decision Letter 1]

3 Jan 2024

Dear Dr. Teng,

Thank you very much for submitting your manuscript "An Epigenomic Fingerprint of Human Cancers by Landscape Interrogation of Super Enhancers at the Constituent Level" for consideration at PLOS Computational Biology.

As with all papers reviewed by the journal, your manuscript was reviewed by members of the editorial board and by several independent reviewers. In light of the reviews (below this email), we would like to invite the resubmission of a significantly-revised version that takes into account the reviewers' comments.

We cannot make any decision about publication until we have seen the revised manuscript and your response to the reviewers' comments. Your revised manuscript is also likely to be sent to reviewers for further evaluation.

Sincerely,

Ilya Ioshikhes

Section Editor

PLOS Computational Biology

Ilya Ioshikhes

Section Editor

PLOS Computational Biology
---

## [Decision Letter · Decision Letter 2]

24 Jan 2024

Dear Dr. Teng,

Thank you very much for submitting your manuscript "An Epigenomic Fingerprint of Human Cancers by Landscape Interrogation of Super Enhancers at the Constituent Level" for consideration at PLOS Computational Biology. As with all papers reviewed by the journal, your manuscript was reviewed by members of the editorial board and by several independent reviewers. The reviewers appreciated the attention to an important topic. Based on the reviews, we are likely to accept this manuscript for publication, providing that you modify the manuscript according to the review recommendations.

Sincerely,

Ilya Ioshikhes

Section Editor

PLOS Computational Biology

Ilya Ioshikhes

Section Editor

PLOS Computational Biology

Reviewer's Responses to Questions

**Comments to the Authors: **

Reviewer #1: The authors have substantially revised the manuscript and addressed my comments. The revised version describes this computational study in a more comprehensive manner. The demonstration of the results in the revised figures has been improved. The findings are validated and significantly highlighted in the revised version of the manuscript. I believe that this work is interesting and provides critical information for understanding super-enhancers assembly and in vivo function. In addition, the work strengthens the resolution/sensitivity, according to which in-depth investigations on both the above parameters can be conducted. It also provides valuable computational strategies/packages that can be utilized from the field. 

Minor Comments

[1] The authors have to describe in more detail the results of figure S7 and figure S8 in the corresponding legends.

Reviewer #2: I appreciate the authors' diligence in addressing the concerns raised during the review process. The revisions made have significantly strengthened the manuscript. Overall, I find the changes satisfactory, and the paper now stands as a more robust contribution to the field.

**Have the authors made all data and (if applicable) computational code underlying the findings in their manuscript fully available?**

Reviewer #1: Yes

Reviewer #2: Yes

PLOS authors have the option to publish the peer review history of their article (what does this mean?). If published, this will include your full peer review and any attached files.

Reviewer #1: No

Reviewer #2: No

Figure Files:

Data Requirements:

Reproducibility:

References:

---

## [Decision Letter · Decision Letter 3]

30 Jan 2024

Dear Dr. Teng,

We are pleased to inform you that your manuscript 'An Epigenomic Fingerprint of Human Cancers by Landscape Interrogation of Super Enhancers at the Constituent Level' has been provisionally accepted for publication in PLOS Computational Biology.

Best regards,

Ilya Ioshikhes

Section Editor

PLOS Computational Biology

Ilya Ioshikhes

Section Editor

PLOS Computational Biology

Reviewer's Responses to Questions

**Comments to the Authors: **

Reviewer #1: The authors have substantially revised the manuscript and addressed my points and questions. 

The revised manuscript is well-written.

I have no other comments.

**Have the authors made all data and (if applicable) computational code underlying the findings in their manuscript fully available?**

Reviewer #1: Yes

PLOS authors have the option to publish the peer review history of their article (what does this mean?). If published, this will include your full peer review and any attached files.

Reviewer #1: No

---

## [Editor Report · Acceptance letter]

5 Feb 2024

PCOMPBIOL-D-23-01108R3 

An Epigenomic Fingerprint of Human Cancers by Landscape Interrogation of Super Enhancers at the Constituent Level

Dear Dr Teng,

I am pleased to inform you that your manuscript has been formally accepted for publication in PLOS Computational Biology. Your manuscript is now with our production department and you will be notified of the publication date in due course.

With kind regards,

Lilla Horvath
